# A workflow for sizing oligomeric biomolecules based on cryo single molecule localization microscopy

**Magdalena C. Schneider**[1]☯, **Roger Telschow**[2]☯, **Gwenael Mercier**[2], **Montserrat López-Martinez**[1], **Otmar Scherzer**[2]*, **Gerhard J. Schütz**[1]*

**1** Institute of Applied Physics, TU Wien, Vienna, Austria, **2** Faculty of Mathematics, University of Vienna, Vienna, Austria

☯ These authors contributed equally to this work.
* otmar.scherzer@univie.ac.at (OS); schuetz@iap.tuwien.ac.at (GJS)

**Data Availability Statement:** All relevant data are within the manuscript and its Supporting information files.

## Abstract

Single molecule localization microscopy (SMLM) has enormous potential for resolving subcellular structures below the diffraction limit of light microscopy: Localization precision in the low digit nanometer regime has been shown to be achievable. In order to record localization microscopy data, however, sample fixation is inevitable to prevent molecular motion during the rather long recording times of minutes up to hours. Eventually, it turns out that preservation of the sample's ultrastructure during fixation becomes the limiting factor. We propose here a workflow for data analysis, which is based on SMLM performed at cryogenic temperatures. Since molecular dipoles of the fluorophores are fixed at low temperatures, such an approach offers the possibility to use the orientation of the dipole as an additional information for image analysis. In particular, assignment of localizations to individual dye molecules becomes possible with high reliability. We quantitatively characterized the new approach based on the analysis of simulated oligomeric structures. Side lengths can be determined with a relative error of less than 1% for tetramers with a nominal side length of 5 nm, even if the assumed localization precision for single molecules is more than 2 nm.

## Introduction

In the last decade, super-resolution microscopy techniques have paved the way for resolving cellular structures in unprecedented detail [1]. The assembly of biomolecules at the nanoscale plays a crucial role in their functionality and hence, is key to our understanding of cellular processes. In particular, the technique of single molecule localization microscopy (SMLM) appears well suited for structural biology, as it is based on localization coordinates of individual molecules rather than on pixelated images of recorded fluorescence intensities. In SMLM, dye molecules are linked to the biomolecule of interest and imaged under conditions, where only a small subset of dye molecules is visible at any time-point. From the movies containing thousands of images of the very same region, one can determine the positions of these dye

**Funding:** OS: F6807-N36, FWF, https://www.fwf.ac.at/ OS: I3661-N27, FWF, https://www.fwf.ac.at/ GJS: F6809-N36, FWF, https://www.fwf.ac.at/ The funders had no role in study design, data collection and analysis, decision to publish, or preparation of the manuscript.

molecules very accurately down to a precision of a few nanometers [2], which allows for establishing localization maps.

The increased spatial resolution of SMLM, however, comes at the cost of temporal resolution, as image acquisition takes several minutes up to hours. Thorough sample fixation is thus a crucial prerequisite for high resolution SMLM recordings, as any residual diffusion of molecules [3] will lead to distortions of the obtained localization maps. Since such residual motion is likely uncorrelated within the sample, it cannot be corrected by standard drift correction methods. Importantly, the chosen fixation method further needs to conserve the ultrastructure of the sample under investigation, which is typically not the case using chemical fixatives [4]. Novel cryo-fixation approaches [5] combined with SMLM at cryogenic temperatures (cryo-SMLM) [6–8] promise to resolve both points, thereby opening up SMLM to questions from structural biology.

Two aspects of SMLM, however, hamper the direct ultrastructural interpretation of localization maps: On the one hand, insufficient labeling and/or detection efficiency leads to undercounting; on the other hand, multiple detections of individual molecules result in overcounting [9]. Therefore, some parts of a particular biomolecular structure may not be visible at all, while other parts may be heavily overrepresented.

In principle, particle averaging approaches allow for circumventing the issue of statistical distortions in SMLM. Similarly to single particle reconstruction methods used in cryo-electron microscopy (cryo-EM), hundreds to thousands of identical copies of the same particle are imaged, and subsequently combined to yield an averaged *super*-particle [10, 11]. In case of unknown structures template-free registration methods have to be employed. Two possible approaches are pyramid registration, where particles are registered pairwise in consecutive steps [12], or all-to-all registration, where all particles are registered to all others simultaneously [13]. Any knowledge of particle symmetry may be included in the registration process in order to increase the quality of the reconstruction [13]. To improve the registration process under realistic imaging conditions, the Bachttacharyya distance allows to account for missing labels, different number of localizations of individual molecules and anisotropic localization uncertainty [12, 13]. In addition, for accurate reconstruction of semi-flexible structures, Shi et al. recently suggested an approach for deformed alignment [14]. Note that up to now these approaches were successfully applied only in case of rather large structures with sizes of tens of nanometers [14], or imaging conditions yielding tens to hundreds of localizations per label site [13]. Quite often, however, the cell-biological context of an experiment is in conflict with these requirements, in many cases impeding particle reconstruction. In such cases, template-based registration methods may recover superresolution analysis, or provide superior results. In principle, template-based registration allows to register the point sets acquired from each particle onto the template. In a pioneering study, Szymborska et al. [15] used a circular template to study the arrangement of molecules in the nuclear pore complex (NPC), allowing the determination of its radius with a precision of 0.1 nm. More elaborate analysis employing the eight-fold symmetry of the NPC allowed to analyze the single-molecule labeling efficiency [16] or reconstruct a more detailed view of the NPC structure [12].

As an alternative to coordinate-based registration, reconstruction can be performed based on algorithms developed for cryo-EM data. In this case, the obtained localization maps first need to be converted to localization images (e.g. based on localization densities or localization uncertainties), since EM-algorithms expect continuous intensity distributions instead of a list of coordinates. Using this approach combined with imaging at cryogenic temperatures, Weisenburger et al. reported a resolution on the Ångström scale for imaging of the GtCitA Pasc domain dimer and the streptavidin homotetramer [8]. Of note, imaging modalities of EM and

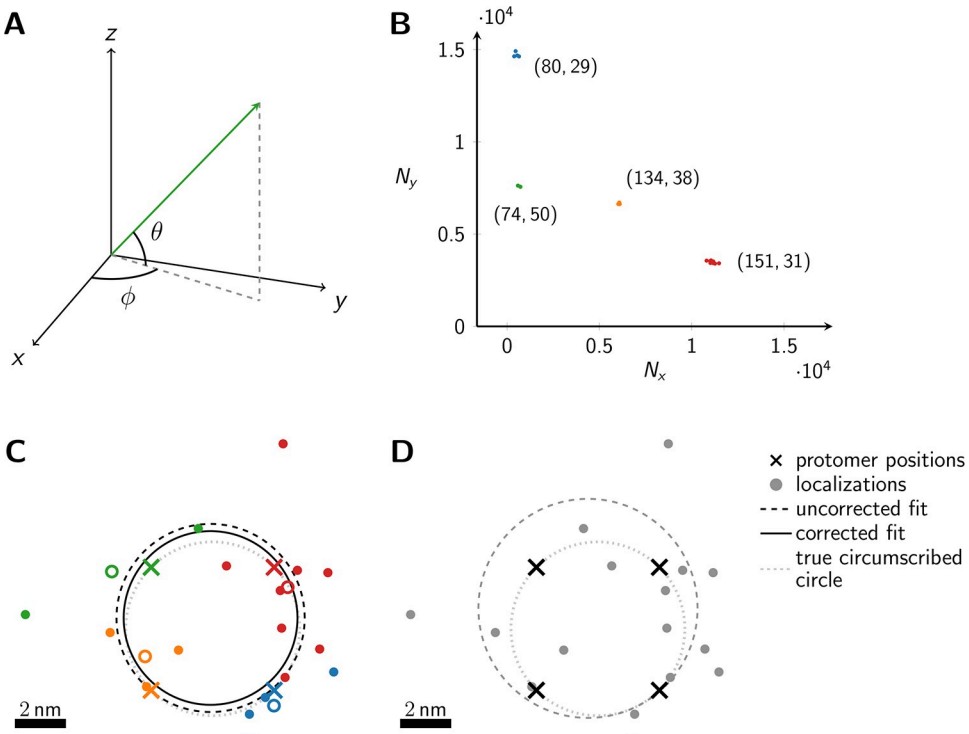

**Fig 1. Schematic representation of the method.** (A) The dipole orientation of a fluorophore is shown as green arrow, which is defined by the azimuthal angle $\varphi$ and the elevation angle $\theta$. Without loss of generality, the optical axis shall be parallel to the z-axis. The fluorophore shall be excited alternately with linearly polarized light of polarization direction along the x- and y- axis. (B) The expected single molecule brightness values for x- and y-polarized light ($N_x$ and $N_y$, respectively) are shown for four exemplary dipole orientations, assuming $N_{max} = 20000$. Localizations corresponding to the same fluorophore are plotted in the same color. Annotated numbers indicate ($\phi$, $\theta$). (C) Exemplary localization map as it could be obtained for a tetramer with a side length of 5 nm. The actual positions of the dye molecules are shown as crosses, the circumscribed circle around the actual tetramer as dotted line. The assignment shown in panel B was used to group localizations, indicated by the same color code. The average position of each localization group is plotted as open dot. The dashed and solid lines show the uncorrected and corrected fitting results for the circumscribed circle (Eqs (10) and (39)). (D) Fitting results of the same localization map as in panel C, but without assignment of localizations to individual dye molecules, yields much worse results (dashed line).

SMLM differ quite substantially and EM-algorithms might not fully account for SMLM specifics.

Currently, however, it is difficult to assign localizations to specific dye molecules. We propose here a new approach for the analysis of oligomeric protein complexes, which is tailored to the conditions of cryo-SMLM. Measuring at cryogenic temperatures has two key advantages, which shall be exploited here: first, it ensures supreme fixation and conservation of the sample's ultrastructure [5]; second, also rotational diffusion of the fluorophores' excitation and emission dipoles during illumination is prevented at least over time-scales of hours [8]. The second aspect allows for establishing a unique characteristic for each dye molecule, based on the orientation of its dipole moment at the time point of freezing. In this paper, we propose to infer this characteristic from imaging sequences, in which samples are alternately excited with linearly polarized light with polarization vectors rotated by 90˚ (Fig 1A). Thereby, assignment of localizations to individual molecules becomes possible, which substantially enhances fitting results. We showcase the performance of the approach by determining the size of regular oligomeric structures, based on the analysis of thousands of simulated oligomers.

## Results

In this manuscript, we consider the analysis of oligomeric protein structures consisting of $n$ protomers, which can be represented by regular polygons consisting of $n$ corners. If not mentioned otherwise, we consider tetramers with a side length of 5 nm. Each protomer shall be labeled by exactly one dye molecule. This can be achieved experimentally, e.g. by using tags or unnatural amino acids as labels [17, 18]. The aim of our study is to develop a template-based analysis approach to determine the distances between individual protomers, by making use of the correct assignment of localizations to individual protomers. The latter shall be enabled by exploiting the linear dichroism observable in the signal molecule brightness, when fixed dipoles are recorded with linearly polarized light.

At cryogenic temperatures, the dipole orientation of a fluorophore is fixed. When exciting such a fluorophore with linearly polarized light, the absorption probability depends on the scalar product between the fluorophore's dipole orientation and the polarization vector of the excitation light (see Eqs (2) and (3)). Exciting the fluorophore consecutively with light of orthogonal polarization directions parallel to the x- and y-axis, respectively (Fig 1A), yields characteristic brightness changes depending on the fluorophore's dipole orientation. Note that the orientation of the $x$, $y$-coordinate system in the image plane can be arbitrary. Here and in the following, we used an analytical representation of the number of localizations per molecule $m$, which closely reflects experimental data (see Methods/*Simulations*); for convenience, we used here data recorded at room temperature. For the single molecule brightness we considered a maximum number of photons per single molecule signal, $N_{max}$, as it would be recorded if the dye's dipole moment was aligned with the polarization of the excitation light. The actual signals, as they would be recorded for arbitrary dipole angles, were calculated according to Eqs (2) and (3), and were subjected to photon shot noise.

Heydarian and colleagues published a template-free approach to analyze the underlying structure of an unknown oligomer based on the obtained localization maps [13]. For high single molecule localization precision characterized by $N_{max} = 10^5$ photons, the method indeed yields satisfactory results and clearly reveals the tetrameric arrangement of the individual dye molecules (see S1C Fig in S1 File). With decreasing photon numbers and increasing localization error, however, localization maps become more difficult to analyze; eventually at $N_{max} = 10^4$ photons per dye molecule, no substructures can be identified. In our manuscript we propose to additionally include information about the assignment of localizations to the individual dye molecules, which becomes available when performing the experiment at cryogenic temperature. As we will show in the following, this assignment not only allows to tackle challenging imaging conditions at low photon numbers, it also yields highly precise estimates of the oligomer size.

In Fig 1B we plot the signal intensities $N_x$, $N_y$ for the two polarization directions for four exemplary fluorophore dipole orientations, as they could occur for a fully labelled tetramer. In this case, discrimination of the four dye molecules is straightforward, and we can group all localizations that belong to each single molecule (indicated by color in Fig 1B and 1C) (see Methods section *Assignment of Blinks to Specific Molecules*). In principle, brightness values can cover the whole region (S2 Fig in S1 File) confined by $N_x > 0$, $N_y > 0$ and $N_x + N_y < N_{max}$, with a slight dip in the center of the region. We only accepted sufficiently bright signals with $N_x + N_y \geq N_{min}$, which would yield a localization error below a user-defined threshold $\Delta x$ (see Methods section *Simulations* for the relation between $\Delta x$ and $N_{min}$, and S2B Fig in S1 File). Note that the point clouds corresponding to each dye molecule can be elliptically distorted due to differences in the Poisson noise along the $x$- and $y$-axis (see the red point cloud in Fig 1B).

For convenience, we assumed throughout our manuscript the following procedure for determining the localization of single molecule signals: The two recordings corresponding to the two polarization channels are added up, irrespective of the signal intensities in the two channels, and the localizations are determined on the sum image. Considering the situation of fixed dipole moments, a substantial fraction of molecules will show dipoles characterized by an elevation angle close to the optical axis. Such molecules will produce rather faint signals, which yield large localization errors. In consequence, a rather broad distribution of localization errors can be expected. Of note, we assumed here subsequent illumination of the sample with different polarizations but detection of the two corresponding images on the very same region of the camera chip; hence, no image registration problems occur.

In Fig 1C and 1D we show the obtained localization map of the exemplary tetramer, both with (C) and without (D) localization assignment. Apparently, without localization assignment there is no realistic chance to identify any structural organization of the oligomer. To facilitate the analysis, one may include prior knowledge e.g. by assuming the oligomer to be represented by a regular polygonic structure. In this case, all corners of the simulated tetramer would lie on the perimeter of the circumscribed circle. However, even under this assumption, the circular fit does not yield satisfactory results (dashed line in Fig 1D); in this particular case, the size of the tetramer is substantially overestimated. Localization assignment substantially improves the situation (**C**). In this case, all localizations assigned to single dye molecules can be averaged, indicated by colored circles in Fig 1C). Taking these averaged positions as input for the fit yields the circle indicated by the dashed line, which is fairly close to the ground truth (dotted line). Importantly, a circle fit shows an inherent bias towards larger sizes [19] (see Methods Eqs (12) & (34)). This is intuitively plausible, as on average more data points lie outside the circle and hence contribute with a higher statistical weight. Correcting for the bias with Eq (39) yields an improved fit result that is shown by the solid line in Fig 1C.

In the following, we provide a quantitative evaluation of the proposed method; specifically, we assess the estimation of oligomer side length from a large number of recorded identical oligomers. We assume here that the oligomers shall be sufficiently separated from each other, so that a standard 2D clustering algorithm can be applied in order to group localizations belonging to individual oligomers. Such clustering algorithms can be found e.g. in refs. [20, 21]. As first step, we group the localizations of each oligomer based on the obtained intensities $N_x$ and $N_y$. As an eligibility criterion, all oligomers which yield $n$ distinct groups of localizations are taken for further analysis, all others are neglected. This criterion particularly rejects scenarios, where two or more groups of localizations overlap and hence would be interpreted as one spurious position at the weighted average of the detected localizations. If not mentioned otherwise, we assumed full labelling of all protomers. In Fig 2A we analyzed the assignment process (gray) and the eligibility criterion (black) for different single molecule brightness levels $N_{max}$. With increasing brightness, we observed an increasing percentage of oligomers for which all localizations were assigned correctly to the individual protomers. The reason for this is the reduced spread of the brightness clusters in the $N_x$—$N_y$ representation, which improves the performance of the applied clustering algorithm. Along a similar line, also the fraction of eligible oligomers increases with $N_{max}$, partly due to improved assignment, partly due to the reduced influence of the detection threshold (S2 Fig in S1 File). We also analyzed the fraction of eligible oligomers which contained incorrectly assigned localizations, yielding negligible contributions (dashed line in Fig 2A).

Secondly, for each group of localizations we calculate their mean position, which are used to fit the circle that minimizes Eq (10); the fits are performed for all oligomers separately. From the fit results we determine the corrected radii $\hat{R}$ using Eq (39) in order to calculate the

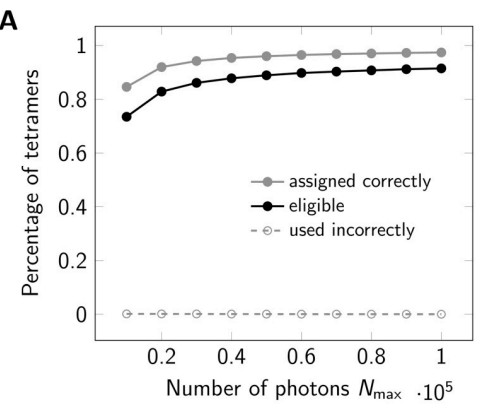
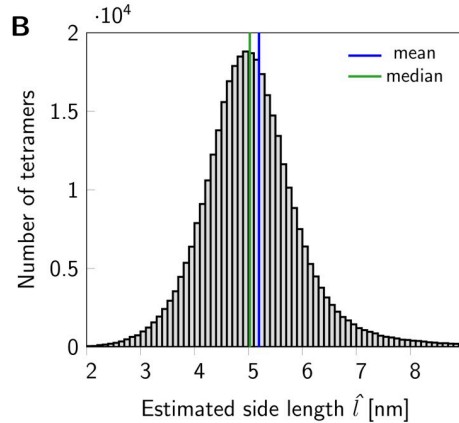

**Fig 2. Statistics for a set of oligomers.** (A) Assignment of localizations to individual dye molecules. The maximum number of photons $N_{max}$ emitted from a fluorophore was varied from 10000 to 100000. The gray line indicates the percentage of tetramers, for which all localizations were assigned to the correct dye molecule. A tetramer was considered eligible for further analysis, if its localizations were assigned into four groups. The black line shows the percentage of eligible tetramers. For each data point a data set of 500000 tetramers (side length 5 nm) was simulated. (B) Histogram of estimated tetramer side length for a data set of 500000 tetramers with nominal side length of 5 nm. Localizations were assigned to individual dye molecules. A total number of 367328 tetramers were eligible and further analyzed. For circle fitting, term (10) was minimized. For side length estimation we used Eq (39). Analysis of the histogram yields a mean of 5.1959 nm (blue line) and a median of 5.0256 nm (green line). Values larger than 9 nm were cut off for display only.

$n$-mer side lengths via $\hat{l} = 2\hat{R}\sin(\pi/n)$. Exemplary fitting results for a simulated data set of 500000 tetramers with a side length of 5 nm are shown in Fig 2B. As this distribution is slightly positively skewed, it seems reasonable to consider the median of the obtained histogram as an estimator of the underlying tetramer side length. Indeed, for this particular case, the median (blue line) outperforms the mean (red line).

We next estimated the influence of the number of simulated oligomers available for the analysis. The results for both a maximum photon number of $N_{max} = 10^4$ and $N_{max} = 10^5$ photons are shown in S3A Fig in S1 File. Here and in subsequent plots, we quantified errors by calculating

$$\varepsilon_l = \left| \frac{\hat{l} - l}{l} \right| \tag{1}$$

where $\hat{l}$ and $l$ denote the determined and nominal side length, respectively. The side length estimation gives rather robust values, which are independent of the number of analyzed oligomers (S3A Fig in S1 File). A marginal bias towards too large or too small values was observed for $N_{max} = 10^4$ and $N_{max} = 10^5$ photons, respectively. As expected, the standard error of the median decreases with increasing number of oligomers (S3B Fig in S1 File). For all subsequent plots we used a total number of 500000 simulated oligomers for the analysis. In this plot, we further compared results obtained from taking the median or the mean values of the individual data sets; comparison shows a much better performance of the median, which was hence taken in all subsequent analyses.

We further analyzed the dependence of $\varepsilon_l$ on the obtained photon numbers by varying the maximal number of photons $N_{max}$ from $10^4$ to $10^5$ (Fig 3A). Note that in this figure, we show a symmetric logarithmic plot, which shows positive and negative relative errors on the positive and negative y-axis, respectively. Relative errors $|\varepsilon_l| < 10^{-3}$ are shown on a linear scale. The median generally gives very precise results with relative errors below 5 ‰, corresponding to

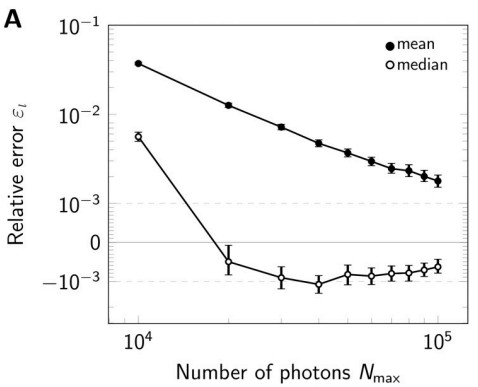
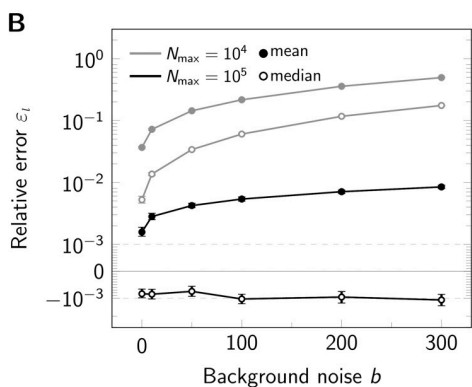

**Fig 3. Influence of signal brightness and noise.** Relative error $\varepsilon_l$ for estimation of tetramer side length upon variation of the maximum photon number $N_{\max}$ (A) and background noise (B) shown in a symmetric logarithmic plot. For each data point 500000 tetramers (side length 5 nm) were simulated. In both panels we compared the analysis via the mean (full symbols) and median (open symbols). Positive and negative relative errors represent overestimation and underestimation, respectively. In panel A we assumed zero background noise $b$, in panel B we considered $N_{\max} = 10^4$ photons (gray lines) and $N_{\max} = 10^5$ (black lines). Error bars indicate the 95% confidence intervals.

0.025 nm. For large photon numbers, $N_{\max} \geq 2 \cdot 10^4$, the side length is slightly underestimated (indicated by red color). Again, the mean estimator performs less well (full symbols). Of note, the average localization error for single molecules $\Delta x$ would yield 2.30 and 0.78 nm for $N_{\max} = 10^4$ and $N_{\max} = 10^5$, respectively.

Up to now, we did not consider background noise for the analysis. A real life experiment, however, inevitably contains contributions from camera noise and sample background noise. The main consequences of including noise in the analysis are increased localization errors. We investigated the influence of background noise on the side length estimation by increasing its standard deviation up to $b = 300$ photon counts, which would be an exceptionally high value for cellular background (Fig 3B and S4 Fig in S1 File). Background noise mainly impacted the results for low photon numbers, where its relative contribution is higher. For high photon numbers, background only had a slight effect on the results.

An important issue with any fluorescence labeling technique is labeling efficiency, leaving some of the protomers within an oligomeric structure undetectable. Experimentally, this may be due to incomplete maturation of fluorescent proteins, prebleaching of dye molecules, or incomplete conjugation of the dye to the protomer. Generally, incomplete labeling compromises registration methods. However, in cases where the template is known and assignment of localizations to individual protomers is possible, one may filter the data and use only oligomers with correct number of dyes $n$ for analysis. To asses the effects of incomplete labeling on our method, we varied the effective labeling efficiency and quantified the eligibility of oligomers. As expected, reduced labeling efficiency massively reduces the number of eligible oligomers (Fig 4A). Importantly, however, the labeling efficiency does not have a large influence on the side length estimation (Fig 4B), only the standard error of the median increases with decreasing labeling efficiency due to the reduced number of eligible oligomers (S5 Fig in S1 File).

We next were interested in the performance of our method for extremely small oligomers. When varying the side length between 10 and 1 nm, we made an interesting observation: While relative errors $\varepsilon_l$ were negligible for side length $l \geq 5$ nm, errors increased strongly at short side lengths, yielding an overestimation of the oligomer size up to a factor of 2 (Fig 5). Relative errors $\varepsilon_l$ were negligible for side length $l \geq 2$ nm and $l \geq 5$ nm for $N_{\max} = 10^5$ and $10^4$ photons, respectively (Fig 5). Errors increased strongly, however, at shorter side lengths,

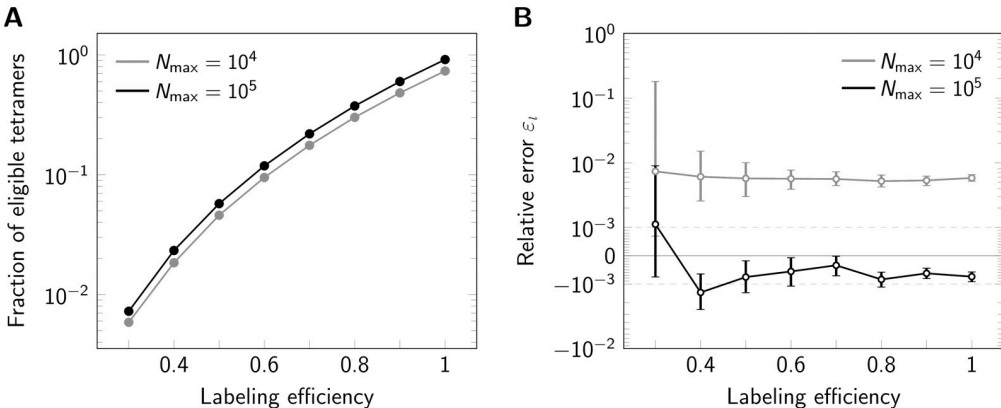

**Fig 4. Influence of labeling efficiency.** For each data point 500000 tetramers (side length 5 nm) were simulated, assuming $N_{max} = 10^4$ photons (gray line) or $N_{max} = 10^5$ (black line). (A) Percentage of eligible tetramers. (B) Relative error $\varepsilon_l$ shown in a symmetric logarithmic plot. Positive and negative relative errors represent overestimation and underestimation, respectively. Error bars indicate the 95% confidence intervals.

yielding an overestimation of the oligomer size up to a factor of 2, likely reflecting increasingly unstable fit results in case of high single molecule localization errors.

Further, we investigated the performance of our method for tri-, tetra-, penta- and hexamers, i.e. oligomers consisting of $n$ = 3, 4, 5, 6 protomers (Fig 6). All oligomers were simulated as regular polygons. For this, we set the radius of each oligomer type to the fixed value of 4 nm. This leads to different side lengths for each oligomer type. The resulting relative error $\varepsilon_l$ of the fitting procedure is shown in (Fig 6B). For both $N_{max} = 10^4$ photons and $N_{max} = 10^5$ photons,

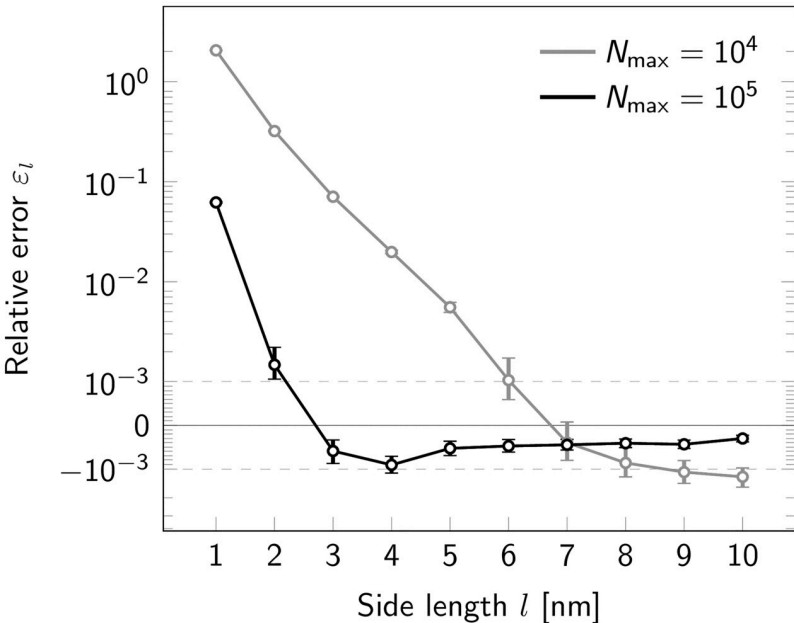

**Fig 5. Influence of oligomer side length.** Relative error $\varepsilon_l$ for estimation of tetramer side length upon variation of the nominal side length $l$ shown in a symmetric logarithmic plot. Results are shown both for a maximum photon number $N_{max} = 10^4$ (gray) and $N_{max} = 10^5$ (black). Positive and negative relative errors represent overestimation and underestimation, respectively. For each data point a data set of 500000 tetramers was simulated. Error bars indicate the 95% confidence intervals.

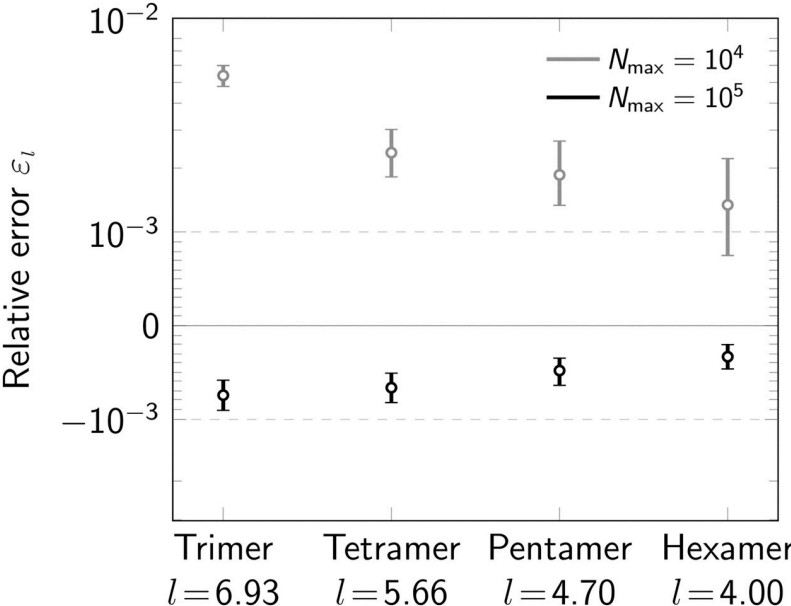

**Fig 6. Influence of oligomerization degree.** Relative error $\varepsilon_l$ for varying degree of oligomerization ($n = 3, 4, 5, 6$) shown in a symmetric logarithmic plot. Data was simulated both for a maximum number of photons $N_{max} = 10^4$ (gray) and $N_{max} = 10^5$ (black). Positive and negative relative errors represent overestimation and underestimation, respectively. For each data point 500000 oligomers were simulated. The radius of the circumscribed circle of the oligomer was set to 4 nm. The resulting side length for each oligomer type is indicated on the $x$-axis. Error bars indicate the 95% confidence intervals.

we observed improved performance with increasing degree of oligomerization. The main reason for this is an increased number of localizations for higher $n$. The number of eligible oligomers is somewhat reduced for increasing $n$ due to increased ambiguities in the localization assignments, and a higher likelihood for missing one of the corners (S6 Fig in S1 File). Importantly, for virtually all simulations we observed very small errors $\varepsilon_l \ll 10^{-2}$.

In a realistic scenario, it may be difficult to ensure coplanarity between the plane of focus and the plane of orientation of the oligomeric structure. We were hence interested to what extent a tilt of tetramers out of the focal plane influences the results. Fig 7 shows that up to 10 degrees tilt the relative errors stay below 1%. Surprisingly, even massive tilts of 40 degrees only lead to a 10% underestimation of the obtained tetramer size.

Finally, we were interested in the performance of our method with respect to runtime (S7 Fig in S1 File). For this, we compared the analysis of different numbers of tetramers on a standard personal computer (see Methods). As input we used localization maps, which were already assigned to individual oligomers. Analysis of 500000 tetramers, as used throughout this manuscript, takes approximately three minutes. As expected, the runtime scales linearly with the number of tetramers, which can become a massive advantage for the analysis of large data-sets compared to template-free methods.

## Discussion

In this manuscript, we describe a workflow for the quantitative analysis of regular oligomeric structures based on single molecule localization microscopy data, that were obtained with polarization-sensitive cryo-fluorescence microscopy. Performing experiments at cryogenic temperatures has a strong advantage over room-temperature measurements, as it solves the

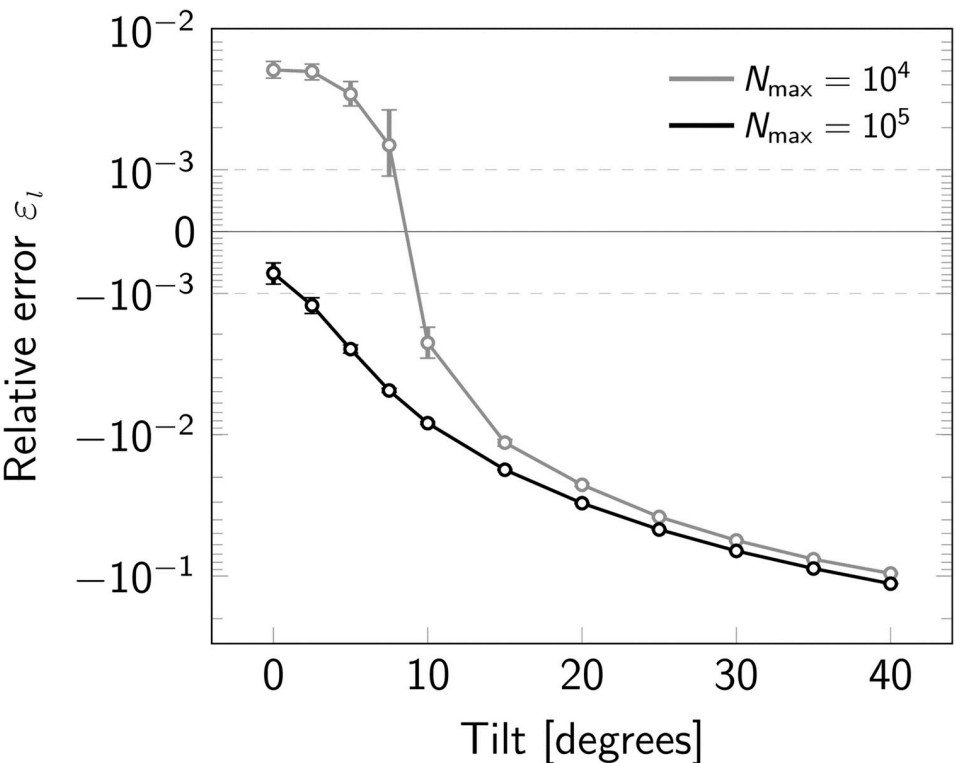

**Fig 7. Influence of a tilt between the focal plane and the oligomerization plane.** Relative error $\varepsilon_l$ for varying tilt of the oligomerization plane shown in a symmetric logarithmic plot. Data was simulated both for a maximum number of photons $N_{max} = 10^4$ (gray) and $N_{max} = 10^5$ (black). Overestimation of oligomer sizes is indicated by positive, underestimation by negative relative errors. For each data point 500000 oligomers were simulated. The radius of the circumscribed circle of the oligomer was set to 4 nm. Error bars indicate the 95% confidence intervals.

fixation problem. Standard fixation methods using chemical fixatives often do not preserve the ultrastructure of the sample [4], and even show residual mobility of biomolecules [3]. The problem becomes massive when SMLM shall be applied to questions from structural biology, where structure sizes down to a few nanometers shall be resolved. In contrast, cryo-fixation is considered as the gold standard and hardly affects the ultrastructure of the sample even below nanometer length scales [5].

Measuring at cryogenic temperatures further offers the possibility to exploit polarization effects due to the fixation of the fluorophore's transition dipole. This allows to assign localizations to the individual dye molecules via their characteristic brightness upon excitation with differently polarized light. On top of that, also differences in the local environment of each fluorophore may additionally accentuate the recorded brightness values, thereby further improving discrimination. In principle, this enables the identification of partially labeled oligomers, which hence can be rejected from the analysis. A further advantage of cryogenic measurements is reduced photobleaching kinetics. In practice, one may hence expect even more precise estimates of oligomer sizes due to a higher number of localizations recorded per molecule.

A few requirements need to be fulfilled in order to fully capitalize on the strength of the method:

- The fluorophores should be located in the focal plane. Due to the fixed orientation of the dipoles, the corresponding PSF will generally be tilted against the optical axis. Even slight

defocusing may hence substantially displace the obtained localizations from the true fluorophore position [22]. Azimuthal filtering [23] or polarization-resolved imaging [24] have been described as solutions to obviate this effect.

- The number of protomers per oligomer should be known. This requirement ensures that only correct, i.e. fully labeled, oligomers are taken for the analysis. While for a substantial number of proteins the degree of oligomerization is known, many interesting cases lack information on the oligomerization. In principle, this information can be extracted from the SMLM data by taking the maximum of the number of localization clusters per oligomer.

- The labeling efficiency should be close to one. The higher the labeling efficiency, the larger the fraction of eligible oligomers (see Fig 4A). In consequence, less experiments would be required to achieve the same quality of the results. Importantly, lower labeling efficiencies do not introduce a bias in the obtained oligomer size, as contributions from incompletely labeled molecules would be rejected before the analysis (see Fig 4B).

- The individual chromophores on each oligomer need to be mutually independent. Dyes in close proximity of a few nanometers may well exhibit coupling between their singlet and triplet levels [25], thereby affecting each other's blinking rates. In the worst case, point-spread functions between different dyes would overlap. As long as reverse intersystem crossing processes are sufficiently slow, however, they can be filtered out in the respective single molecule trajectories based on abrupt jumps in the single molecule orientation. Of note, such effects were not observed in the pioneering study by Weisenburger et al. [8].

- The oligomeric structure should be a regular polygon. While the analysis of irregular polygonal structures is in principle feasible, such a treatment goes beyond the scope of this manuscript.

- The population should be homogeneous. Heterogeneous sample compositions are a challenging scenario for any particle averaging approach. In case of heterogeneities in the degree of oligomerization, our approach would yield the size of the oligomers of highest degree present in the sample.

- The mutual distance between different oligomers should be large. To avoid localizations overlapping between different oligomers, it is critical to ensure that the mutual distance $d$ of two neighboring oligomers is much larger than the spread of localizations belonging to one oligomer, i.e. $d \gg R + \Delta x$. This can be achieved by reducing expression levels and/or by increasing the localization precision.

- Oligomerization should occur only in a plane perpendicular to the optical axis. Two scenarios may be discriminated: First, the oligomerization plane may be tilted against the optical axis. An example would be oligomerization of proteins within the plasma membrane, which is not perfectly flat. In consequence, the obtained structures are distorted (Fig 7). Up to 40 degrees tilt angles, our approach yields oligomer sizes with surprisingly high precision. In more extreme cases, however, one may revert to alternative strategies. For example, a straightforward solution would be the rejection of oligomers, which show localization maps deviating from a regular polygon. Slightly distorted structures could still be accounted for by including a deformation matrix in the model [14]. Secondly, biomolecules may also oligomerize in three dimensions. In this case, tomographic approaches [8] may be preferential.

A straightforward application of our method would be the study of the protomer arrangement within oligomeric structures. Quite often it is not clear in which orientation protomers

are assembled or how particular domains of the protein are arranged. If site-specifically labeled protomers are available, the resulting side length would depend on the position of the label: Labels facing towards the inside of the oligomeric structure would yield smaller side lengths than labels facing the outside of the oligomer. Positioning labels on specific sites of the protein hence allows for unravelling the protomer orientation. Similar approaches proved to be successful for the analysis of larger structures such as nuclear pore complexes [15] or endocytic sites [26].

In order to fully exploit the potential of our method it is critical to choose the labeling strategy wisely: Labels should be sufficiently small to report on the actual position of the target site on the protein, and exactly one dye molecule should be linked to the target site. These constraints disqualify fluorescently labeled antibodies. Appropriate possibilities include small tags [17] and unnatural amino acids [18]. In principle, also switchable fluorescent proteins can be used for the analysis of oligomeric structures which are large compared to the size of the fluorescent protein.

## Conclusion

Taken together, we have presented and quantitatively characterized a method for polarization-sensitive cryo-SMLM. We found remarkable precision for the determination of the side length of regular oligomeric structures with relative errors of less than 1%, which would be of sufficient quality to ascribe subunit positions in multi-protein complexes. We believe that our method provides a good basis for opening up structural biology applications to cryo-SMLM approaches.

## Methods

### Simulations

First, we simulated the positions of the protomers. For this, $n$ protomers were assigned to each $n$-mer ($n$ = 3, 4, 5, 6). Individual protomers belonging to one oligomer were arranged around the oligomer's center position in the shape of a regular polygon with fixed side length, but random in-plane orientation. If not specified otherwise, we simulated $\mathcal{N}_{\mathrm{oligo}} = 500\,000$ oligomers for each analyzed data set.

Second, each protomer was assumed to be labeled with exactly one dye molecule. In order to account for recordings at cryogenic conditions, a random but fixed dipole orientation was assigned to each dye molecule. The inherent brightness $N_{\mathrm{max}}$ was considered to be the same for all dye molecules.

To simulate blinking, we assigned a random number of detections to each dye molecule, which was drawn from an artificial blinking statistics following a log-normal distribution (as in [27]). The mean of the log-normal distribution was set to 6.4 localizations and the standard deviation to 5 localizations. These values correspond to previously reported blinking characteristics of fluorescent probes under realistic experimental conditions (compare [28]).

Fluorophores were simulated to be excited alternatingly with differently polarized excitation light. The coordinate system was aligned with the orthogonal polarization directions $x$ and $y$, which are orthogonal to the optical axis $z$. The absorption probability of a fluorophore depends on the angle between its dipole orientation and the polarization of the excitation light. Hence, w.l.o.g. the effective number of photons $N_x$, $N_y$ for the two polarizations of

excitation light can be calculated as

$$N_x = N_{\text{max}} \cos^2(\theta) \cos^2(\phi) \tag{2}$$

$$N_y = N_{\text{max}} \cos^2(\theta) \sin^2(\phi) \tag{3}$$

where $\theta$ and $\phi$ are the elevation and azimuth angle of the fluorophore's dipole relative to the $x$–axis, respectively (see Fig 1), and $N_{\text{max}}$ the number of photons emitted if dipole orientation coincides with the polarization vector of the excitation light. For all simulations, we assumed random distributions of $\theta$ and $\phi$ on a sphere. The resulting probability density for detecting $(N_x, N_y)$ photons is given by (see Note 1 in S1 File)

$$\rho_{\text{phot.}}(N_x, N_y) = \begin{cases} \frac{1}{2\pi} \left( N_{\text{max}} N_x N_y (N_{\text{max}} - N_x - N_y) \right)^{-\frac{1}{2}} & \text{for } N_x, N_y \geq 0, \\ & N_x + N_y \leq N_{\text{max}}, \\ 0 & \text{otherwise.} \end{cases} \tag{4}$$

Photon shot noise was included by drawing the observed number of photons from Poisson distributions with mean $N_x$ and $N_y$, respectively.

The error in intensity estimation was distributed according to a normal distribution with mean 0 and variance $(\Delta N)^2$. The variance $(\Delta N)^2$ was set to the best possible variance of an unbiased estimator, which corresponds to the Cramér-Rao lower bound (CRLB) and is given as follows [29]:

$$\langle (\Delta N)^2 \rangle = N \left( 1 + 4\tau + \sqrt{\frac{\tau}{14(1 + 2\tau)}} \right), \quad \text{with } \tau = \frac{2\pi b(\sigma_{\text{PSF}}^2 + a^2/12)}{Na^2}, \tag{5}$$

where $a$ is the pixel size, $b$ the background noise, $N$ the signal photon count (i.e. $N_x$, $N_y$) and $\sigma_{\text{PSF}}$ the standard deviation of the point-spread function (PSF). If not mentioned otherwise, background noise was set to $b = 0$. We assumed a pixel size of 100 nm and a standard deviation of the PSF of 160 nm.

Determination of the single molecule positions was assumed to be performed based on the combined images acquired by excitation with differently polarized light. The total intensity was calculated as $N_{\text{total}} = N_x + N_y$. The uncertainty of the localization procedure is hence given as [29]:

$$\langle (\Delta x)^2 \rangle = \frac{\sigma_{\text{PSF}}^2 + a^2/12}{N_{\text{total}}} \left( 1 + 4\tau + \sqrt{\frac{2\tau}{1 + 4\tau}} \right). \tag{6}$$

As the background noise of the two individual frames combines, $b$ was replaced by $\sqrt{2} \cdot b$ in the calculation of $\tau$. Localization coordinates were displaced from the true protomer position by adding a random localization error according to the localization precision $\Delta x$. Any detections with a localization precision below 10 nm were discarded. Together with given values of background noise $b$, pixel size $a$ and the standard deviation of the PSF $\sigma_{\text{PSF}}$ this defines a minimum number of required photons to detect a single molecule signal $N_{\text{min}}$.

In order to simulate tilted tetramers, without loss of generality we assumed a tilt around the x-axis. To this end, we transformed the y-coordinates of the single molecule positions according to $y' = y \cdot \cos(\alpha)$, where $\alpha$ denotes the tilt angle of the oligomerization plane with respect to the focal plane.

## Mathematical analysis

In this mathematical part, we will use the following notation. We assume that all oligomers are equilateral polygons and have the same number of corners $n$. We will need to distinguish between the different dye molecules constituting an oligomer, which we will index by $i \in \{1, \cdots, n\}$, and different localizations corresponding to dye molecule $i$, which we will index by $j \in \{1, \cdots, m_i\}$, where $m_i$ specifies the total number of localizations of dye molecule $i$. The position of the individual dye molecule will be denoted by

$$\mathbf{p}_i = (x_{\mathrm{p}}^{(i)}, y_{\mathrm{p}}^{(i)}) \tag{7}$$

whereas we will write for the positions of the blinks $\mathbf{b}_j^{(i)} = (x_j^{(i)}, y_j^{(i)})$. A superscript $T$ as in $(x, y)^T$ denotes the transpose of a vector or matrix and $(\cdot)_x$ and $(\cdot)_y$ yields the $x$ and $y$-component of a vector, respectively. The value $\mathbb{E}(\rho)$ denotes the expectation value of a random variable $\rho$, and its variance is defined by $\mathbb{E}([\rho - \mathbb{E}(\rho)]^2)$. Moreover, the addition of a hat, as in $\hat{\rho}$, is used for the estimator of a certain random variable. Note that an estimator $\hat{\rho}$ is called unbiased if $\mathbb{E}(\hat{\rho}) = \mathbb{E}(\rho)$. Notably, there might be the situation where the estimator is biased, in which case $\mathbb{E}(\hat{\rho}) = \mathbb{E}(\rho) + \mathbb{B}(\hat{\rho})$ where $\mathbb{B}(\hat{\rho})$ is called the bias.

Throughout the manuscript we make use of the following nomenclature: $R$, $L$ denote the ground truth radius and side length, respectively, of a regular polygon of $n$ corners, which are related via $L = 2R \sin(\pi/n)$. For each oligomer $i$ with given dipole orientations of the dye molecules, the variables $\hat{r}_i$ and $\hat{l}_i$ denote the estimators for radius and side length, as they are obtained from the circle fit (described in section *Method for Minimization*). In particular, they are not corrected for the fitting bias (described in section *Identification of the Bias*). Note that $\hat{r}_i$ and $\hat{l}_i$ are randomly distributed due to the presence of localization errors. The variables $\hat{R}$ and $\hat{L}$ denote the bias-corrected estimators for radius and side length of all oligomers $\mathcal{N}_{\mathrm{oligo}}$, which happen to be eligible for analysis (see section on *Assignment of Blinks to Specific Molecules*). As discussed in the main text of this paper, we calculated the estimator $\hat{R}$ via the mean or median value of all $\hat{r}_i$.

**Identification of individual oligomeric structures.**   First of all, we assume that the individual oligomers as well as the corresponding measured localizations are well separated from the ones of each other oligomer. That is, measurements of different oligomers do not overlap. If that is the case, we are able to cluster the given data spatially in order to identify the localizations belonging to individual oligomers. This can be done effectively with standard two dimensional clustering techniques. We use a straightforward approach. We sort the data and take the differences in coordinates in order to identify the adjacent blinks which are closer than a certain prescribed distance from one another. Every such localizations are then grouped together to one cluster. In the simulations, however, we know which protomer belongs to which oligomer such that we omit this step and use the given information in order to avoid unlikely errors in this regard.

**Assignment of blinks to specific molecules.**   This task is performed by taking advantage of the measured polarization of the dipole. In general, the spatial variance of the distribution of blinks makes a reliable clustering (using only the spatial data) impossible. However, the polarization property discriminates effectively between all molecules in one oligomer, provided the polarization of each protomer is sufficiently far from the one of each other. It has to be noticed that we do not have access to the whole polarization of these molecules, but only to their projection on the illumination plane. Similarly, a sign change in the polarization cannot be detected.

Concretely, we cluster the estimated intensity of the two polarization directions, which is again a clustering in 2D as done above in the spatial domain (if performed). We consider the oligomer to be well resolved if the collection of blinks corresponding to that oligomer can be clustered in $n$ groups, where the distance between two groups has to be larger than a given parameter $\delta$. Empirical tests suggest $\delta_p = 300 + N_{max}/100$ to be a feasible choice for difference in the number of photons in order to assert the localizations properly. In the case the cluster is not well resolved for the polarization, we simply discard it for our further computations. Otherwise, we call the oligomer eligible and proceed to estimate the distance between its individual protomers.

See also Fig 1 for visualization, where one specific example (tetramer) is shown and the corresponding blinks are assigned to their respective protomer. As we can see, a simple spatial clustering of the blinks is not applicable.

**Estimation of distance between single protomers.**   Given the data of one individual oligomer, i.e., the localizations (blinks) contained in one spatial cluster, we are now interested in the distance $l$ between the individual protomers. Since we assume the oligomeric structure to be a regular polygon, the distance between two adjacent protomers is supposed to be constant. That is, assuming that the corners are ordered, for all $i$, we have

$$l = \left| (x_p^{(i)}, y_p^{(i)}) - (x_p^{(i+1)}, y_p^{(i+1)}) \right| = const. \tag{8}$$

For regular polygons, the radius $R$ and the distance $l$ between adjacent protomers are directly related via $l = 2R \sin(\pi/n)$. In order to find an estimation for this distance, we use a geometric circle fit. Intuitively, we could minimize the mean square distance between the data points and the fitted circle, which more precisely means solving

$$\min_{\substack{a,b \in \mathbb{R} \\ R \geq 0}} \sum_{i=1}^{n} \sum_{j=1}^{m_i} \left( \sqrt{\left(x_j^{(i)} - a\right)^2 + \left(y_j^{(i)} - b\right)^2} - R \right)^2. \tag{9}$$

However, this turns out to yield unsatisfactory results, as we can see in Fig 1A. Due to our ability to figure out which blinks belongs to which protomer, we fit the circle to the centers of mass of these individual clusters of blinks (see Fig 1B). That means, instead of the above we actually solve

$$\min_{\substack{a,b \in \mathbb{R} \\ R \geq 0}} \sum_{i=1}^{n} \left( \sqrt{\left(\bar{x}_i - a\right)^2 + \left(\bar{y}_i - b\right)^2} - R \right)^2, \tag{10}$$

where the means are given by

$$\bar{x}_i = \frac{1}{m_i} \sum_{j=1}^{m_i} x_j^{(i)}, \quad \bar{y}_i = \frac{1}{m_i} \sum_{j=1}^{m_i} y_j^{(i)}, \quad i = 1, \ldots, n. \tag{11}$$

Note that the minimization takes place over the center $(a, b) \in \mathbb{R}^2$ of the circle as well as its radius $\mathbb{R} \in [0, +\infty)$. The example the different fits in Fig 1 show that the assignment of the blinks to their respective protomers improves the center as well as the estimation of the radius.

Although the standard (affine fit) least square problem is strictly convex and has a unique solution, this one is not convex and can have several minima which might correspond to unrealistic solutions [30]. Moreover, this procedure will always yield an estimation for the radius of the circle which possesses a certain bias and the radius $R$ of the circle (see [19, 31, 32]). For example, in Fig 2 one finds the histogram of the distribution of the estimated side length for

individual tetramers of 5 nm side length. As we can see, the mean of the distribution is higher than the ground truth and the median, although also overestimated, is closer to the real value.

**Identification of the bias.**   Suppose the localizations of the blinks are identically and independently distributed (iid) normal random variables with zero expectation (centered) and constant variance $\sigma^2$. Depending on these parameters, we define the random variable $r$ which describes the radius of the circle fitting those blinks (by minimizing (10)) and we denote by $\hat{r}$ the estimator of $R$. As described in [19], the bias of this estimator is in this case essentially given by

$$\mathbb{E}(\hat{r} - R) = \mathbb{B}(\hat{r}) \simeq \frac{\sigma^2}{2R} \tag{12}$$

In our setup, however, the variance of the random variables is not constant since there are different deviations for the localizations belonging to each individual protomer due to the polarization of the light. For the moment let us consider one fixed spatial cluster of blinks. That is, the data corresponding to one individual oligomer. For each protomer $\mathbf{p}_i = (x_{\mathrm{p}}^{(i)}, y_{\mathrm{p}}^{(i)})$, $i = 1, \ldots, n$, contained in the oligomeric structure, the coordinates of the blinks $\mathbf{b}_j^{(i)} = (x_j^{(i)}, y_j^{(i)})$, $j = 1, \ldots, m_i$, are mathematically actually realizations of

$$\beta_j^{(i)} := \mathbf{p}_i + \zeta_j^{(i)} = (x_{\mathrm{p}}^{(i)} + \xi_j^{(i)}, y_{\mathrm{p}}^{(i)} + \eta_j^{(i)}) \tag{13}$$

with centered independent identically distributed (iid) normal random variables $\xi_j^{(i)}, \eta_j^{(i)}$ with variance $\sigma_i^2$, i.e., $\xi_j^{(i)}, \eta_j^{(i)} \sim \mathcal{N}(0, \sigma_i^2)$. Let the mean values of these variables be denoted by

$$\bar{\xi}_i = \frac{1}{m_i} \sum_{j=1}^{m_i} \xi_j^{(i)}, \quad \bar{\eta}_i = \frac{1}{m_i} \sum_{j=1}^{m_i} \eta_j^{(i)}. \tag{14}$$

Obviously, we obtain expectations $\mathbb{E}(\bar{\xi}_i) = \mathbb{E}(\bar{\eta}_i) = 0$ and, consequently, the corresponding variances are given by

$$\mathbb{E}(\bar{\xi}_i^2) = \mathbb{E}(\bar{\eta}_i^2) = \frac{\sigma_i^2}{m_i}, \quad i = 1, \ldots, n, \tag{15}$$

since the $\xi_j^{(i)}$ and $\eta_j^{(i)}$ are iid. In order to have an estimation of the position of the protomer $\mathbf{p}_i$ we use the center of mass of the random blinks which is simply given by the mean

$$\hat{\mathbf{p}}_i = \frac{1}{m_i} \sum_{j=1}^{m_i} \beta_j^{(i)} = \mathbf{p}_i + \frac{1}{m_i} \sum_{j=1}^{m_i} \zeta_j^{(i)} = \left(x_{\mathrm{p}}^{(i)} + \bar{\xi}_i, y_{\mathrm{p}}^{(i)} + \bar{\eta}_i\right). \tag{16}$$

For each measured blink $\mathbf{b}_j^{(i)}$ we are given an estimation for its standard deviation, denoted as $s_j^{(i)}$, in $x$ as well as in $y$-direction. That is, the variance in each random coordinate of $\beta_j^{(i)}$, which means the variance of $\xi_j^{(i)}$ and $\eta_j^{(i)}$, can be estimated by the (sample) variance

$$\hat{\sigma}_i^2 = \frac{1}{m_i} \sum_{j=1}^{m_i} (s_j^{(i)})^2. \tag{17}$$

Consequently, the estimator $\hat{\mathbf{p}}_i$ possesses a variance $\mathbb{E}([\hat{\mathbf{p}}_i - \mathbf{p}_i]^2)$ which is simply the variance of the mean coordinate deviations $\bar{\xi}_i$ and $\bar{\eta}_i$ as given in Eq (15). Eventually, this can be

estimated by

$$\hat{z}_i = \frac{\hat{\sigma}_i^2}{m_i} = \frac{1}{m_i^2} \sum_{j=1}^{m_i} (s_j^{(i)})^2, \quad i = 1, \ldots, n.$$

(18)

Due to the fact that the variance is different for every $(x_p^{(i)}, y_p^{(i)})$, we cannot simply use the bias given in (12). Consulting the considerations in [19], we obtain the following. In our case of a geometric circle fit, we have the parameter vector $\Theta = (a, b, R)$ and denote its estimator by $\hat{\Theta} = (\hat{a}, \hat{b}, \hat{R})$. Moreover, we can write

$$x_p^{(i)} = a + R \cos \varphi_i, \quad y_p^{(i)} = b + R \sin \varphi_i$$

(19)

with given angles $\varphi_i \in [0, 2\pi[$ for $i = 1, \ldots, n$. Then, we denote

$$u_i = \cos \varphi_i = \frac{x_p^{(i)} - a}{R}, \quad v_i = \sin \varphi_i = \frac{y_p^{(i)} - b}{R}$$

(20)

such that $u_i^2 + v_i^2 = 1$ for all $i = 1, \ldots, n$ and we define

$$\mathbf{W} = \begin{pmatrix} u_1 & v_1 & 1 \\ \vdots & \vdots & \vdots \\ u_n & v_n & 1 \end{pmatrix}, \quad \mathbf{U} = \text{diag}(u_1, \ldots, u_n), \quad \mathbf{V} = \text{diag}(v_1, \ldots, v_n).$$

(21)

Hence, with $\tilde{\Theta} = \hat{\Theta} - \Theta$, the first order expansion (in $\sigma$) of the minimization problem for the geometric circle fit can be written as

$$\mathbf{W} = \Theta \approx \mathbf{U}\xi + \mathbf{V}\eta + \mathcal{O}_{\mathbb{P}}(\sigma^2)$$

(22)

with $\xi = (\bar{\bar{\xi}}_1, \ldots, \bar{\bar{\xi}}_N)^T$ and $\eta = (\bar{\eta}_1, \ldots, \bar{\eta}_N)^T$ which has the solution

$$\tilde{\Theta} = (\mathbf{W}^T \mathbf{W})^{-1} \mathbf{W}^T (\mathbf{U}\xi + \mathbf{V}\eta)$$

(23)

when the $\mathcal{O}_{\mathbb{P}}(\sigma^2)$ terms are neglected. Then the covariance of the statistical error is (to the leading order) given by

$$\mathbb{E}(\tilde{\Theta}\tilde{\Theta}^T) = (\mathbf{W}^T \mathbf{W})^{-1} \mathbf{W}^T \mathbf{Z} \mathbf{W} (\mathbf{W}^T \mathbf{W})^{-1},$$

(24)

where

$$\mathbf{Z} = \mathbb{E}\big((\mathbf{U}\xi + \mathbf{V}\eta)(\xi^T \mathbf{U} + \eta^T \mathbf{V})\big) = \text{diag}\left(\frac{\sigma_1^2}{M_1}, \ldots, \frac{\sigma_n^2}{M_n}\right).$$

(25)

Note that in the case of a constant variance $\sigma^2$ and single measurements per point (i.e., without averaging as in (14)), the latter simplifies to $\sigma^2 \mathbf{I}$ such that $\mathbb{E}(\tilde{\Theta}\tilde{\Theta}^T) = \sigma^2 (\mathbf{W}^T \mathbf{W})^{-1}$ which coincides with the result obtained in [19].

To improve the estimation of the bias, we use the second order Taylor expansion of (10) for our parameters $\Theta = (a, b, R)$, which we write for the sake of brevity

$$\hat{a} = a + \tilde{a}_1 + \tilde{a}_2 + \mathcal{O}_{\mathbb{P}}(\sigma^3) \tag{26}$$

$$\hat{b} = b + \tilde{b}_1 + \tilde{b}_2 + \mathcal{O}_{\mathbb{P}}(\sigma^3) \tag{27}$$

$$\hat{R} = R + \tilde{R}_1 + \tilde{R}_2 + \mathcal{O}_{\mathbb{P}}(\sigma^3) \tag{28}$$

where the first order terms $\tilde{a}_1, \tilde{b}_1, \tilde{R}_1$ are linear combinations of the variables $\bar{\bar{\xi}}_i, \bar{\eta}_i$ and already known as they are contained in the Matrix given in (24). That means, we need to know the second order terms $\tilde{a}_2, \tilde{b}_2, \tilde{R}_2$ which are quadratic forms of the latter random variables. After expansion and simplification in (10), this yields another minimization problem which reads

$$\sum_{i=1}^{n} \left( u_i \tilde{a}_2 + v_i \tilde{b}_2 + \tilde{R}_2 - h_i \right)^2 \longrightarrow \min, \tag{29}$$

where the values $h_i$ for $i = 1, \ldots, n$ are given by

$$\begin{aligned} h_i = \quad & u_i \left( \bar{\bar{\xi}}_i - \tilde{a}_1 \right) + v_i \left( \bar{\eta}_i - \tilde{b}_1 \right) - \tilde{R}_1 + \frac{v_i^2}{2R} \left( \bar{\bar{\xi}}_i - \tilde{a}_1 \right)^2 \\ & + \frac{u_i^2}{2R} \left( \bar{\eta}_i - \tilde{b}_1 \right)^2 + \frac{u_i v_i}{R} \left( \bar{\bar{\xi}}_i - \tilde{a}_1 \right) \left( \bar{\eta}_i - \tilde{b}_1 \right). \end{aligned} \tag{30}$$

Setting $\mathbf{h} = (h_1, \ldots, h_n)^T$ and $\Phi_2 = (\tilde{a}_2, \tilde{b}_2, \tilde{R}_2)^T$, we find the solution

$$\Phi_2 = (\mathbf{W}^T \mathbf{W})^{-1} \mathbf{W}^T \mathbf{h}, \tag{31}$$

where $\mathcal{O}_{\mathbb{P}}(\sigma^3)$ terms are neglected. Hence, we obtain

$$\mathbb{E}(\tilde{\Theta}) = \mathbb{E}(\Phi_2) = (\mathbf{W}^T \mathbf{W})^{-1} \mathbf{W}^T \mathbb{E}(\mathbf{h}). \tag{32}$$

In contrast to the case of a constant variance the latter is now a three-component vector which in general does not contain any zeros. That means, even the estimation of the center of the circle is going to be biased. On the other hand, this perfectly makes sense since some of the points to be fitted are simply measured more accurately than others, which indeed has an impact on the choice of the center. However, since the center is not of particular importance for our computations, we ignore the fact that the center is biased and only correct the estimation of the radius. Without loss of generality, we set $\varphi_i = \frac{i 2\pi}{n}$ for $i = 1, \ldots, n$ since we know that the underlying oligomeric structure is an equilateral polygon. We can always rotate our coordinate system or reindex such that this is fulfilled. Eventually, this gives us the matrices $\mathbf{U}, \mathbf{V}$ as well as $\mathbf{W}$. Moreover, we replace $\mathbf{Z}$ in (24) by its estimator

$$\hat{\mathbf{Z}} = \mathrm{diag}\left( \hat{z}_1^2, \ldots, \hat{z}_n^2 \right) = \mathrm{diag}\left( \frac{\hat{\sigma}_1^2}{M_1}, \ldots, \frac{\hat{\sigma}_n^2}{M_n} \right) \tag{33}$$

according to Eqs (17) and (18) in order to approximate the matrix $\mathbb{E}(\tilde{\Theta} \tilde{\Theta}^T)$ given by (24). With these values we are able to compute the vector $\mathbf{h}$ as well as $\mathbb{E}(\tilde{\Theta})$ which gives us the desired bias for the particular oligomeric setup (tetramer, pentamer, hexamer etc). The bias

relevant for our computations writes

$$\mathbb{E}(\hat{r} - R) = \frac{1}{R}\left(\frac{1}{2}\frac{\sum_{i=1}^{n}\hat{z}_i}{n} + \frac{1}{n}\frac{\sum_{i=1}^{n}\hat{z}_i}{n}\right) = \frac{1}{R}\left(\frac{1}{2n}\sum_{i=1}^{n}\hat{z}_i + \frac{1}{n^2}\sum_{i=1}^{n}\hat{z}_i\right) \tag{34}$$

Comparing again with the situation in [19] of $M$ single iid measurements with variance $\sigma^2$, we obtain in analogy to Eq (12) the bias

$$\mathbb{E}(\hat{r} - R) = \frac{1}{R}\left(\frac{1}{2}\frac{M\sigma^2}{M} + \frac{1}{M}\frac{M\sigma^2}{M}\right) = \frac{\sigma^2}{2R} + \frac{\sigma^2}{RM}, \tag{35}$$

where the second summand is neglected in [19] since it vanishes asymptotically for $M$ tending to infinity (the paper [19] expands asymptotically with a number of measurments growing to infinity). Hence, the first summand, which does not depend on the number of measurements taken, is the so called essential bias. In our case, however, the second summand in (34) must not be neglected since our $n$ is small (usually $n = 4, 5, 6$). On the other hand, it is easy to see that the $\hat{z}_i$ become smaller, the more measurements $m_i$ we have for a single protomer such that the entire bias tends to zero for $m_i \to \infty$, $i = 1, \ldots, n$. Therefore, more measurements lead to more accurate estimations of the radius and, in turn, the length of the edges of the oligomers. This is not the case in the setup described in [19]. The effect of the bias is shown in Figs 1 and 2.

An exact solution of Eq (34), considering $\mathbb{E}(\hat{r} - R) = \mathbb{E}(\hat{r}) - R$, is given by

$$\mathbb{E}(\hat{r}) = R + \frac{\sum_{i=1}^{n}\hat{z}_i}{R}\left(\frac{1}{2n} + \frac{1}{n^2}\right) \tag{36}$$

such that we eventually find

$$R_{1,2} = \frac{\mathbb{E}(\hat{r})}{2} \pm \sqrt{\left(\frac{\mathbb{E}(\hat{r})}{2}\right)^2 - \left(\frac{1}{2n} + \frac{1}{n^2}\right)\sum_{i=1}^{n}\hat{z}_i}. \tag{37}$$

In case of $R > \frac{\sum_{i=1}^{n}\hat{z}_i}{R}\left(\frac{1}{2n} + \frac{1}{n^2}\right)$, the correct solution is given by

$$R = \frac{\mathbb{E}(\hat{r})}{2} + \sqrt{\left(\frac{\mathbb{E}(\hat{r})}{2}\right)^2 - \left(\frac{1}{2n} + \frac{1}{n^2}\right)\sum_{i=1}^{n}\hat{z}_i}. \tag{38}$$

Since $\mathbb{E}(\hat{r})$ is experimentally not accessible, we can replace it by the random variable $\hat{r}$. Hence, a bias-corrected estimator $\hat{r}_{\text{corr}}$ for the oligomer radius is given by

$$\hat{r}_{\text{corr}} = \frac{\hat{r}}{2} + \sqrt{\left(\frac{\hat{r}}{2}\right)^2 - \left(\frac{1}{2n} + \frac{1}{n^2}\right)\sum_{i=1}^{n}\hat{z}_i}. \tag{39}$$

An estimator of the radius based on the analysis of $\mathcal{N}_{\text{oligo}}$ oligomers is obtain via

$$\hat{R} = \mathbb{E}(\hat{r}_{\text{corr}}) \text{ or } \hat{R} = \text{median}(\hat{r}_{\text{corr}}). \tag{40}$$

If not indicated otherwise, the median was taken for analysis.

**Method for minimization.** In order to solve (10), we have to provide an initial guess ($a_0$, $b_0$) for the center as well as $r_0$ for the radius. While the minimization is not too sensitive to the guessed radius, the initial coordinates for the center ought to be not too far away from the ground truth. For that purpose, we use the center of mass as an initial guess for the center point. If we would not assign the different blinks to their individual protomers, that is the

situation of $M$ iid measurements, we simply compute the overall center of mass of all blinks in that particular spatial cluster. In our notation, that means with $M = \sum_{i=1}^{n} m_i$ we would have the center

$$\left(\tilde{a}_0, \tilde{b}_0\right) = \frac{1}{nM} \sum_{i=1}^{n} \sum_{j=1}^{m_i} \left(x_j^{(i)}, y_j^{(i)}\right)$$

as our initial guess for the minimization. However, we are able to assign blinks to their respective protomer such that we actually obtain

$$(a_0, b_0) = \frac{1}{n} \sum_{i=1}^{n} \frac{1}{m_i} \sum_{j=1}^{m_i} \left(x_j^{(i)}, y_j^{(i)}\right) \tag{41}$$

as the center of mass of our oligomer, which will in general differ from the former. For the radius we simply choose a value $R_0$ which is sufficiently small such that $R_0 < R_{\text{truth}}$. Hence, with the centers of mass of the individual clusters of blinks (clustered in intensity), which are the estimations for the positions of the single protomers, given by the means $\bar{x}_i$ and $\bar{y}_i$ (see Eq 11), we eventually obtain the initial guess

$$(a_0, b_0) = \frac{1}{n} \sum_{i=1}^{n} (\bar{x}_i, \bar{y}_i) \,. \tag{42}$$

The method we use in order to perform the minimization in (10) is based on the Levenberg-Marquardt algorithm (LM, see [33, 34]). The LM method is an iterative optimization algorithm to solve non-linear least square problems (like the one above). In most applications it tends to be slower than a classical Gauß-Newton approach but, on the other hand, it is more robust. Essentially, LM is a Gauß-Newton ansatz which incorporates a regularization term which forces a decay of function values in the process. To be more specific, let us introduce the following notions. Let the function $f : \mathbb{R}^5 \to \mathbb{R}$ be defined by

$$f(x, y, a, b, R) = \sqrt{(x-a)^2 + (y-b)^2} - R$$

such that the sum in (10) is given by

$$S(a, b, R) = \sum_{i=1}^{n} [f(\bar{x}_i, \bar{y}_i, a, b) - R]^2 \,. \tag{43}$$

In each iteration of the LM method the initial guess $(a_0, b_0, R_0)$ is now replaced by values $(a_k, b_k, R_k)^T = (a_{k-1}, b_{k-1}, R_{k-1})^T + \mathbf{d}$ with $\mathbf{d} = (d_x, d_y, d_R)^T$ and $k \in \mathbb{N}$. Let us further define the matrix $\mathbf{D}$ whose rows consist of the derivatives

$$\mathbf{D}_i = \left(\frac{\partial f(\bar{x}_i, \bar{y}_i, a, b, R)}{\partial a}, \frac{\partial f(\bar{x}_i, \bar{y}_i, a, b, R)}{\partial b}, -1\right), \quad i = 1, \ldots, n \,. \tag{44}$$

Hence, for sufficiently small $\mathbf{d}$, we can use the linearized approximation

$$f(\bar{x}_i, \bar{y}_i, x_k, y_k, R_k) \approx f(\bar{x}_i, \bar{y}_i, x_{k-1}, y_{k-1}, R_{k-1}) + \mathbf{D}_i \mathbf{d} \,,$$

which means the sum in Eq (43) can be approximately written as

$$S(a_k, b_k, R_k) = S(a_{k-1} + d_x, b_{k-1} + d_y, R_{k-1} + d_R) \tag{45}$$

$$\approx \sum_{i=0}^{n} [f(\bar{x}_i, \bar{y}_i, a_{k-1}, b_{k-1}, R_{k-1}) + \mathbf{D}_j \mathbf{d}]^2. \tag{46}$$

Taking the derivative with respect to $d_x$, $d_y$, and $d_r$, setting the gradient equal to zero, and using matrix/vector notation, we now obtain with the vectors $\mathbf{f} = (f(\bar{x}_i, \bar{y}_i, a_{k-1}, b_{k-1}, R_{k-1}))_{i=1,\ldots,n}^T$ that

$$\mathbf{D}^T \mathbf{D} \mathbf{d} = -\mathbf{D}^T \mathbf{f} \tag{47}$$

has to hold for the minimum of $S$. In the case of the LM algorithm this system of equations is numerically stabilized by the addition of the identity matrix $\mathbf{I}$ with factor $\lambda > 0$, which means (47) is replaced by

$$(\mathbf{D}^T \mathbf{D} + \lambda \mathbf{I}) \mathbf{d} = -\mathbf{D}^T \mathbf{f} \tag{48}$$

The regularization parameter $\lambda$ can be changed in every iteration in order to speed up the convergence of the algorithm.

Since we can estimate the localization precision (standard deviation in nm) from the measured intensity for each individual blink of one protomer, we are able to discard low quality data. Hence, one could ignore measurements whose localization precision is lower than certain threshold in order to improve results.

**Calculation of error bars.** All error bars were calculated based on 1000 bootstrap samples, which were drawn from the individual data sets, and represent the 95% confidence intervals of the mean (or median).

**Runtime analysis.** For analysis of the runtime shown in S7 Fig in S1 File we used a standard personal computer model XPS 15 9570 with an Intel Core i7-8750H processor.

## Supporting information

**S1 File. Supplementary figures and supplementary note.**
(PDF)

**S2 File. The latest version of the software is available on GitHub at the following link:**
**https://github.com/schuetzgroup/sizingOligomersCryoSMLM.**
(ZIP)

## Acknowledgments

We thank Hamidreza Heydarian and Bernd Rieger for assistance with the code to generate S1 Fig in S1 File.

## Author Contributions

**Conceptualization:** Magdalena C. Schneider, Roger Telschow, Gwenael Mercier, Montserrat López-Martinez, Otmar Scherzer, Gerhard J. Schütz.

**Formal analysis:** Magdalena C. Schneider, Roger Telschow, Gwenael Mercier.

**Funding acquisition:** Otmar Scherzer, Gerhard J. Schütz.

**Investigation:** Magdalena C. Schneider, Roger Telschow, Gwenael Mercier.

**Methodology:** Magdalena C. Schneider, Roger Telschow, Gwenael Mercier, Otmar Scherzer, Gerhard J. Schütz.

**Resources:** Otmar Scherzer, Gerhard J. Schütz.

**Software:** Magdalena C. Schneider, Roger Telschow, Gwenael Mercier.

**Supervision:** Otmar Scherzer, Gerhard J. Schütz.

**Writing – original draft:** Magdalena C. Schneider, Roger Telschow, Gwenael Mercier, Gerhard J. Schütz.

**Writing – review & editing:** Magdalena C. Schneider, Montserrat López-Martinez, Otmar Scherzer, Gerhard J. Schütz.

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
