## [Decision Letter · Decision Letter 0]

20 Nov 2020

PONE-D-20-25964

A workflow for sizing oligomeric biomolecules based on cryo single molecule localization microscopy

PLOS ONE

Dear Dr. Schütz,

Thank you for submitting your manuscript to PLOS ONE. After careful consideration, we feel that it has merit but does not fully meet PLOS ONE’s publication criteria as it currently stands. Therefore, we invite you to submit a revised version of the manuscript that addresses the points raised during the review process.

Please address the comments raised by the reviewers to the best of your ability. Update figures and discussions where appropriate, especially to satisfy concerns related to statistical analysis.

We would appreciate receiving your revised manuscript by Jan 04 2021 11:59PM. If you will need more time than this to complete your revisions, please reply to this message or contact the journal office at plosone@plos.org. Please include the following items when submitting your revised manuscript:

We look forward to receiving your revised manuscript.

Kind regards,

Pushkar P Lele

Academic Editor

PLOS ONE

Journal Requirements:

Reviewers' comments:

Reviewer's Responses to Questions

**Comments to the Author**

1. Is the manuscript technically sound, and do the data support the conclusions?

Reviewer #1: Yes

Reviewer #2: Yes

2. Has the statistical analysis been performed appropriately and rigorously? 

Reviewer #1: I Don't Know

Reviewer #2: Yes

3. Have the authors made all data underlying the findings in their manuscript fully available?

Reviewer #1: Yes

Reviewer #2: Yes

4. Is the manuscript presented in an intelligible fashion and written in standard English?

Reviewer #1: Yes

Reviewer #2: Yes

5. Review Comments to the Author

Reviewer #1: It is an interesting work, I have raised some concerns in the attached review.

Reviewer #2: The authors present a theoretical study how polarization-sensitive cryo-SMLM in combination with particle averaging analysis can be used for super-resolution imaging of regular oligomeric structures with side lengths of a few nanometers. If feasible, the method can be used advantageously for the study of multiprotein complexes and other biologically interesting nanoscale structures. Currently insufficient labeling with large linkage errors prevent the translation of the method into experimental validation but I am personally optimistic that new labeling schemes involving genetic code expansion can pave the way for applications of polarization-sensitive cryo-SMLM. In combination with cryofixation methods, the approach would preserve the ultrastructure even of sensitive protein complexes. The gain in localization assignment to a specific emitter comes from the determination of the unique transition dipole moment of each emitter, which is fixed at cryogenic temperatures.

Taking together, the manuscript describes an interesting approach and useful theoretical study. I have, however, one concern, which the authors should address before publication. I am wondering how energy transfer pathways (energy hopping, single-singlet annihilation, etc.) between nanometer separated dipoles influences or in the worst case scenario prevents the localization of all emitters attached to the regular oligomeric structure. While I believe these processes are of minor importance for distances larger than 5 nm they certainly complicate data analysis for shorter distances.

6. PLOS authors have the option to publish the peer review history of their article (what does this mean?). If published, this will include your full peer review and any attached files.

Reviewer #1: No

Reviewer #2: No

---

## [Author Response · Author response to Decision Letter 0]

18 Dec 2020

We thank both reviewers for their thoughtful inputs to our manuscript. In the revised version we addressed all issues raised. In the attached document, you find a point-by-point reply, including references to the changes in the manuscript. Note that line numbers refer to the manuscript with tracked changes.

---

## [Editor Report · Decision Letter 1]

6 Jan 2021

A workflow for sizing oligomeric biomolecules based on cryo single molecule localization microscopy

PONE-D-20-25964R1

Dear Dr. Schütz,

We’re pleased to inform you that your manuscript has been judged scientifically suitable for publication and will be formally accepted for publication once it meets all outstanding technical requirements.

Kind regards,

Pushkar P Lele

Academic Editor

PLOS ONE

---

## [Editor Report · Acceptance letter]

8 Jan 2021

PONE-D-20-25964R1 

A workflow for sizing oligomeric biomolecules based on cryo single molecule localization microscopy 

Dear Dr. Schütz:

I'm pleased to inform you that your manuscript has been deemed suitable for publication in PLOS ONE. Congratulations! Your manuscript is now with our production department. 

Kind regards, 

on behalf of

Dr. Pushkar P Lele 

Academic Editor

PLOS ONE